# Isotopologues of potassium 2,2,2-trifluoroethoxide for applications in positron emission tomography and beyond

Qunchao Zhao[1], Sanjay Telu [1] ✉, Susovan Jana [1], Cheryl L. Morse [1] & Victor W. Pike [1] ✉

The 2,2,2-trifluoroethoxy group increasingly features in drugs and potential tracers for biomedical imaging with positron emission tomography (PET). Herein, we describe a rapid and transition metal-free conversion of fluoroform with paraformaldehyde into highly reactive potassium 2,2,2-trifluoroethoxide ($CF_3CH_2OK$) and demonstrate robust applications of this synthon in one-pot, two-stage 2,2,2-trifluoroethoxylations of both aromatic and aliphatic precursors. Moreover, we show that these transformations translate easily to fluoroform that has been labeled with either carbon-11 ($t_{1/2} = 20.4$ min) or fluorine-18 ($t_{1/2} = 109.8$ min), so allowing the appendage of complex molecules with a no-carrier-added [11]C- or [18]F- 2,2,2-trifluoroethoxy group. This provides scope to create candidate PET tracers with radioactive and metabolically stable 2,2,2-trifluoroethoxy moieties. We also exemplify syntheses of isotopologues of potassium 2,2,2-trifluoroethoxide and show their utility for stable isotopic labeling which can be of further benefit for drug discovery and development.

Trifluoromethylation finds extensive utility in medicinal chemistry and has led to many drug-like compounds with improved pharmacokinetic and physicochemical properties[1]. Over the past decade, substantial advances have been made in trifluoromethylation methods[2–4]. Fluoroform ($HCF_3$) is a major industrial byproduct and now gains attention as an affordable and atom-efficient source of the trifluoromethyl group ($CF_3$)[5,6]. Strategies for installing a trifluoromethyl group from fluoroform deploy nucleophilic or metal-mediated conversions (Fig. 1A). Nucleophilic trifluoromethylations rely on the deprotonation of fluoroform in the presence of a strong base to generate the reactive trifluoromethyl anion[7], which can subsequently undergo reaction with a wide range of electrophiles[8]. Metal-mediated trifluoromethylation reactions[9] convert fluoroform into a more stable copper(I) ($CuCF_3$[10]) or silver(I) ($AgCF_3$[11]) derivative, which can engage productively in diverse reactions.

Positron emission tomography (PET) is a molecular imaging modality that now plays a vital role in biomedical research, drug discovery, disease staging, and disease diagnosis[12,13]. PET tracers are produced at time of need and the majority are labeled with cyclotron-produced short-lived carbon-11 ($t_{1/2} = 20.4$ min) or fluorine-18 ($t_{1/2} = 109.8$ min)[14–16]. [11]C-Labeled tracers are valuable for studies that demand multiple imaging sessions within the same day, whereas [18]F-labeled tracers may be distributed to remote PET imaging facilities that lack an on-site cyclotron and radiochemistry facility. The production of PET tracers for biomedical applications relies on straightforward radiolabeling strategies. For carbon-11, [11]C]iodomethane and [11]C]methyl triflate are of foremost importance for tracer syntheses through reactions with carbon and heteroatom electrophiles[14–17]. For fluorine-18, nucleophilic substitution on appropriate precursors with [18]F]fluoride is widely used[18,19] (Fig. 1C). However, these and many other methods are limited with regards to the chemotypes that can be labeled and to the possible molecular positions that are open to labeling. Because of tracer metabolism, the position of radiolabel can be a key determinant of PET tracer efficacy[20].

[1]Molecular Imaging Branch, National Institute of Mental Health, National Institutes of Health, 10 Center Drive, Bethesda, MD 20892–1003, USA.
✉e-mail: sanjay.telu@nih.gov; pikev@mail.nih.gov

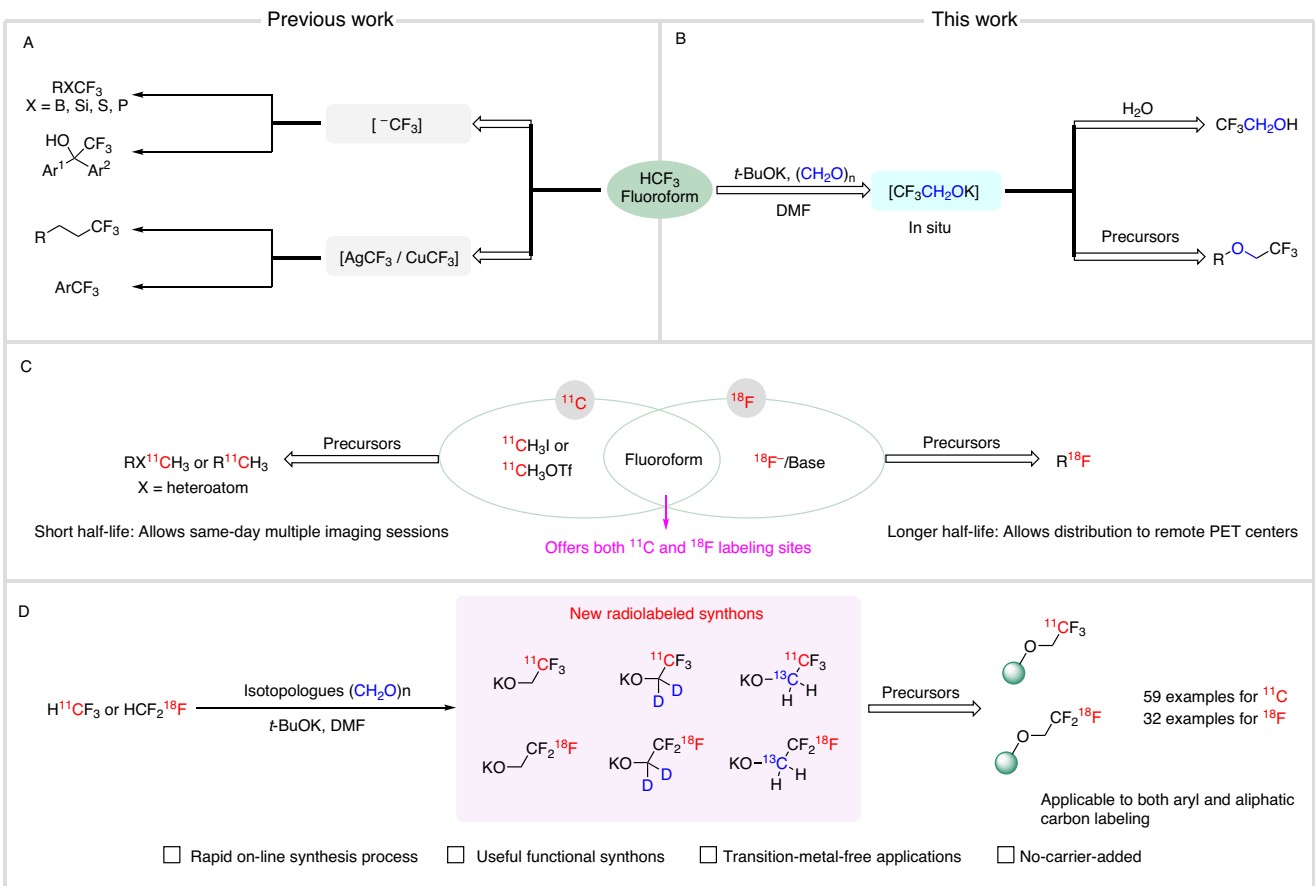

**Fig. 1 | Approaches for transferring a trifluoromethyl moiety from iso-topologues of fluoroform. A** Known approaches. **B** The approaches presented in this work. **C** Major methods for producing PET tracers from [11]C- and [18]F-labeled synthons. **D** The methodology for rapid and efficient radiotrifluoroethoxylations presented in this work.

There is an ongoing need to expand the range of chemotypes that may be considered as potential PET tracers through the development of labeling synthons and methods[21]. Methods to label a single tracer with either carbon-11 or fluorine-18 normally require different non-radioactive precursors and entail major synthesis campaigns that may consume excessive time and resources. A more efficient approach is to take advantage of chemical moieties that contain both carbon-11 and fluorine-18 labeling sites and offer labeling via a single precursor. This strategy can simplify the PET radiotracer development process and more effectively address design requirements. In this respect, [11]C- and [18]F-labeled fluoroforms[22–24] are attractive synthons because they are readily accessible through rapid on-line syntheses as precursors to putatively metabolically stable trifluoromethyl groups (Fig. 1C). Nonetheless, these labeling synthons have been applied almost exclusively to labeling arenes. Application to labeling non-functionalized aliphatic substrates has not been demonstrated.

The 2,2,2-trifluoroethoxy group[25] has notable metabolic stability[26] and moderate lipophilicity and is found in numerous biologically active compounds[27–30]. Several PET radiotracers feature a 2,2,2-trifluoroethoxy moiety (e.g., tracers for COX-1[31,32], tauopathy[33–36], and Huntington aggregates[37]). Installation of a radiolabeled 2,2,2-tri-fluoroethoxy group in such tracers has been approached through either nucleophilic addition of [18F]fluoride to a geminal difluorovinyl precursor[31,33,36,37] or alkylation of a phenoxy precursor with [18F]2,2,2-trifluoroethoxy tosylate[31,35,38]. However, both methods suffer from limited substrate scope and low molar activity (ratio of radioactivity to mass of tracer isotopologues) that may make the derived tracers unsuitable for application[20,39].

Herein, we describe a rapid and transition metal-free conversion of fluoroform into highly reactive potassium 2,2,2-trifluoroethoxide (CF$_3$CH$_2$OK) and demonstrate robust applications of this synthon in one-pot, two-stage 2,2,2-trifluoroethoxylations of both aromatic and aliphatic precursors (Fig. 1B). Moreover, we show that this transformation translates easily to both carbon-11 and fluorine-18 chemistry, so allowing the appendage of complex molecules with [11]C- or [18]F- labeled 2,2,2-trifluoroethoxy groups (Fig. 1D). We also exemplify syntheses of isotopologues of potassium 2,2,2-trifluoroethoxide and show their utility for stable isotopic labeling.

## Results and discussion
### Synthesis and reactivity of potassium 2,2,2-trifluoroethoxide
Nucleophilic addition of fluoroform to aldehydes to produce carbinols is well-known[40]. However, the reaction of fluoroform with the simplest aldehyde, formaldehyde, has not been explored. We considered that this reaction should produce a very useful reagent, 2,2,2-trifluoroeth-oxide. Because of the toxicity and volatility of formaldehyde[41], we decided to explore solid paraformaldehyde as an alternative for producing 2,2,2-trifluoroethoxide. Treatment of fluoroform as limiting reagent with a mixture of paraformaldehyde and t-BuOK in DMF produced 2,2,2-trifluoroethanol in 41% yield after water quench (Supplementary Table 1, entry 1). Under optimal conditions, fluoroform was converted into 2,2,2-trifluoroethanol in 88% yield within 30 min (Supplementary Table 1, entry 13). Increasing the reaction time did not improve the conversion appreciably (Supplementary Table 1, entries 11 and 12). Other bases or solvents adversely affected yield (Supplementary Table 1, entries 14–18). We next tested the reactivity of the

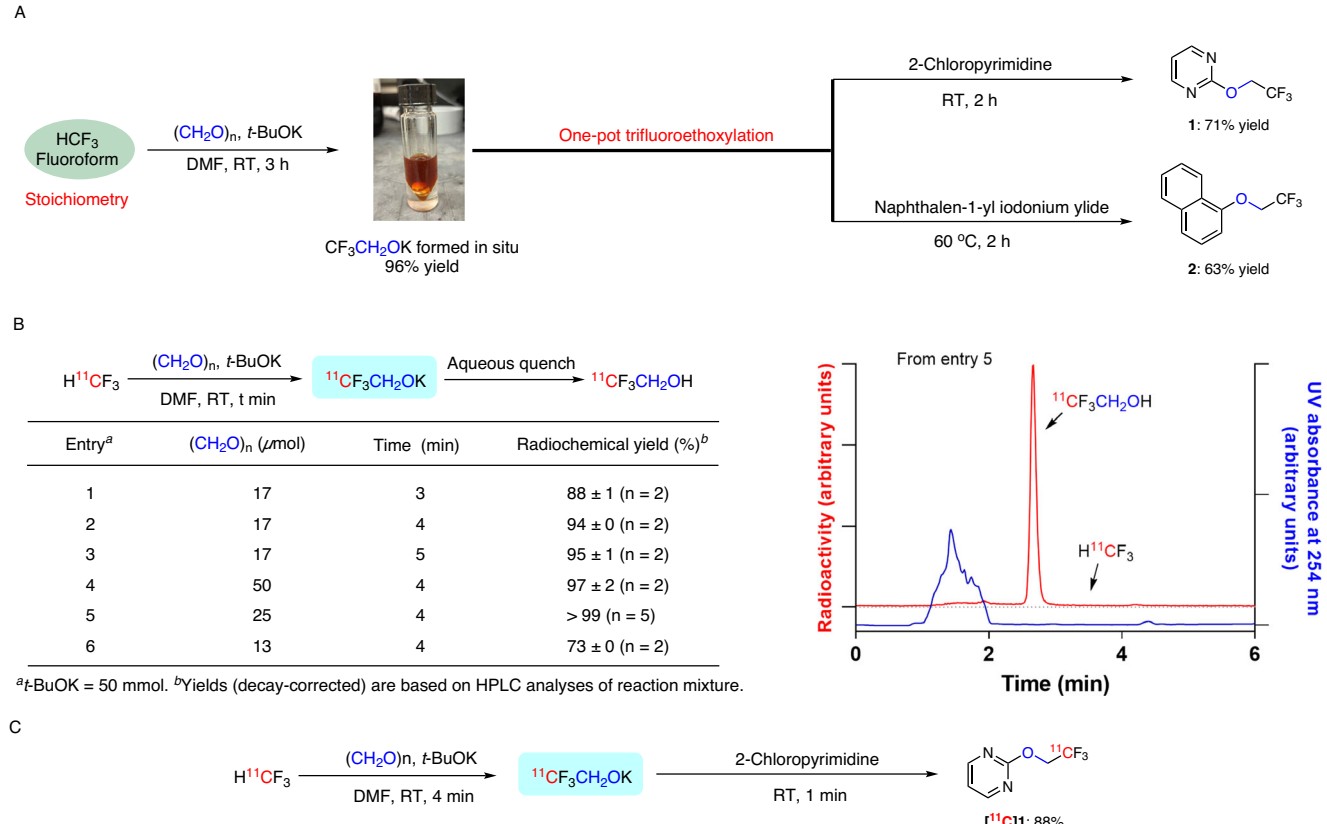

**Fig. 2 | Reactions of fluoroform and [¹¹C]fluoroform. A** One-pot two-stage tri-fluoroethoxylations with fluoroform to give 2,2,2-trifluoroethyl ethers. Yields were determined with ¹⁹F NMR spectroscopy. (For more detail, see Supplementary Table 1 and Supplementary Fig. 1). **B** Optimization of conversion of [¹¹C]fluoroform with paraformaldehyde into [2-¹¹C]2,2,2-trifluoroethanol (left panel) and HPLC chromatogram of the crude reaction mixture from entry 5 (right panel). **C** Example of one-pot, two-stage synthesis of 2-[¹¹C](2,2,2-trifluoroethoxy)pyrimidine.

putative intermediate, potassium 2,2,2-trifluoroethoxide, by treatment with 2-chloropyrimidine. To our delight, reaction proceeded smoothly at room temperature to afford 2-(2,2,2-trifluoroethoxy)pyrimidine[1] in 71% yield within 2 h (Fig. 2A). However, treatment of potassium 2,2,2-trifluoroethoxide with 1-fluoronaphthalene gave only trace 1-(2,2,2-trifluoroethoxy)naphthalene[2], even at elevated temperature. We considered that an aryl hypervalent iodine center in an aryliodonium ylide might show more reactivity. Indeed, treatment of potassium 2,2,2-trifluoroethoxide with naphthalen-1-yl iodonium·(2,2-dimethyl-1,3-dioxane-4,6-dione)ylide at 60 °C produced **2** in 63% yield (Fig. 2A). Thus, gratifyingly, we had succeeded in converting stoichiometric amounts of fluoroform into a useful synthon, potassium 2,2,2-trifluoroethoxide, in high yield and in showing the utility of this synthon for tri-fluoroethoxylation of homoarene and heteroarene under mild conditions. We anticipate that this transition metal-free transformation can find wide applications in fluorine chemistry. Indeed, we readily produced several trifluoroethoxy compounds in high yields by this chemistry as standards for use in the remainder of this study (see Supporting Information, Supplementary methods).

### Synthesis of [¹¹C]potassium 2,2,2-trifluoroethoxide

We next aimed to explore this synthetic methodology for robust broad-scope radio-trifluoroethoxylations as a potential route to PET tracers. For this purpose, we routinely produce [¹¹C]fluoroform by CoF₃-mediated fluorination of cyclotron-produced [¹¹C]methane[22]. Treatment of [¹¹C]fluoroform (37–296 MBq) with a mixture of t-BuOK (50 μmol) and paraformaldehyde (17 μmol) in DMF for only 3 min at room temperature gave an excellent yield (88%) of ¹¹CF₃CH₂OH upon hydrolysis of the putative intermediate (Fig. 2B, entry 1). Increasing the

reaction time to 5 min only slightly increased the yield. A larger quantity of paraformaldehyde (50 μmol) afforded ¹¹CF₃CH₂OH in 97% yield. The yield of ¹¹CF₃CH₂OH was quantitative when [¹¹C]fluoroform was treated with 1: 2 molar mixture of paraformaldehyde and t-BuOK for 4 min (Fig. 2B, entry 5). These reaction conditions were therefore deemed optimal. This success encouraged us to test the efficacy of ¹¹CF₃CH₂OK for introducing the ¹¹C-trifluoromethyl moiety into a wide range of compounds.

### Synthesis of ¹¹C-2,2,2-trifluoroethoxy arenes

First, we examined the reactivity of ¹¹CF₃CH₂OK towards heteroarenes. To our delight, ¹¹CF₃CH₂OK reacted at room temperature to produce a wide range of desired ¹¹C-2,2,2-trifluoroethoxy heteroarenes in moderate to excellent yields within just 1 min (Fig. 3A). Attractive features to emerge from this labeling protocol were: (1) the ¹¹CF₃CH₂OK, can be used without isolation; (2) reaction conditions are mild and rapid; (3) substrate scope is broad, and encompasses pyridines, pyrimidines, pyrazine, thiazoles, triazines, quinolines, and isoquinolines; (4) in addition to halides (F, Cl, and Br), leaving groups such as nitro ([¹¹C]**4**) and methyl sulfone ([¹¹C]**9**) are highly effective; (5) functional group tolerance is high, with aldehyde ([¹¹C]**3**), bromo ([¹¹C]**4**, [¹¹C]**6**), nitrile ([¹¹C]**5**), methoxy ([¹¹C]**7**), and Boc protection ([¹¹C]**9**) all well tolerated. Heteroarenes having 1 to 3 nitrogen atoms ([¹¹C]**1**, [¹¹C]**3**–[¹¹C]**14**) were compatible with the reaction conditions and afforded the desired ¹¹C-labeled products in acceptable yields (21–94%). Furthermore, the late-stage ¹¹C-trifluoroethoxylation of complex biomolecules was highly effective as shown by the labeling of analogs of several drug-like compounds [Imiquimod ([¹¹C]**15**), Milrinone ([¹¹C]**18**)], drug precursors [Erlotinib ([¹¹C]**19**), Canagliflozin ([¹¹C]**20**), Pazopanib ([¹¹C]**21**),

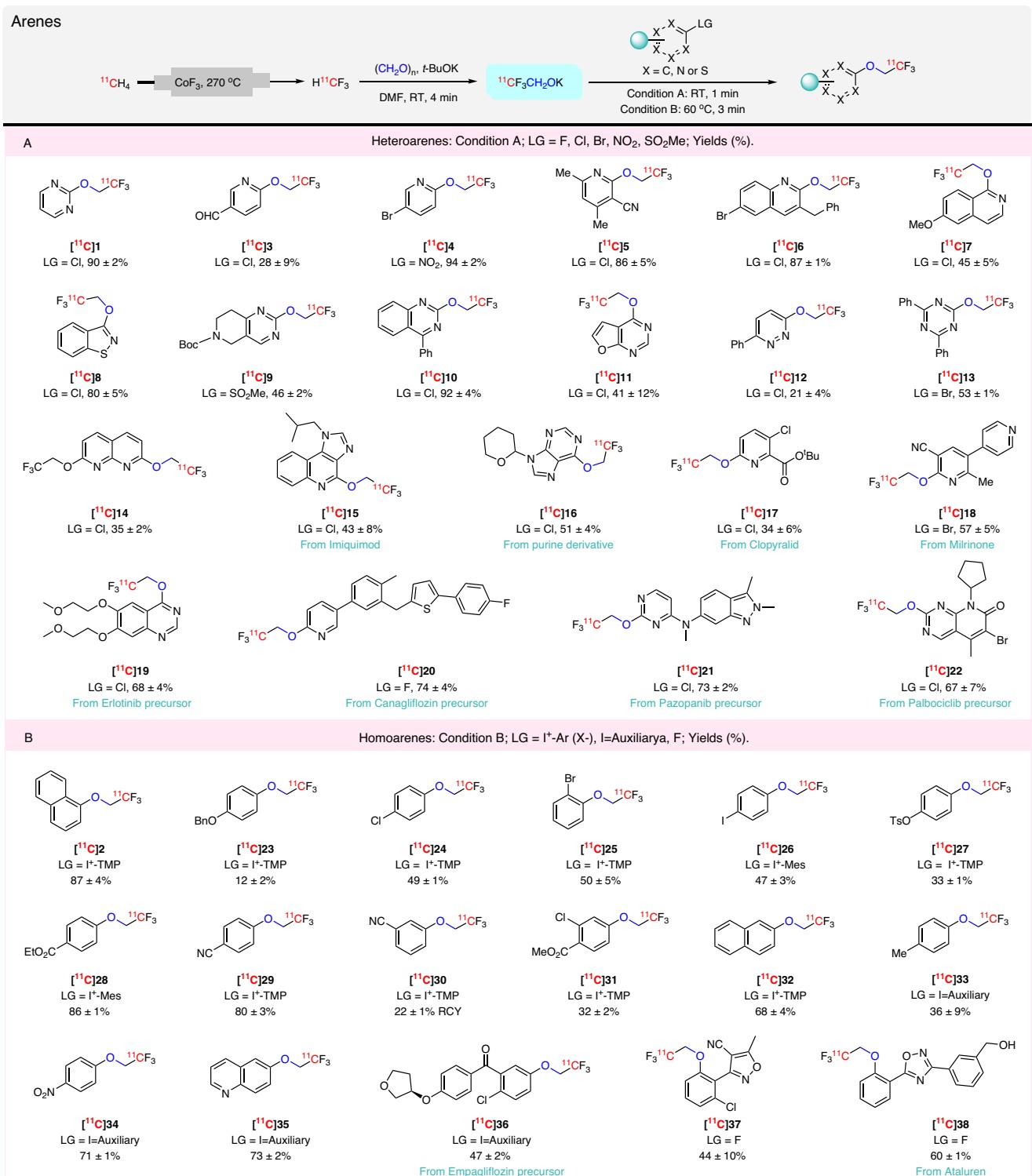

**Fig. 3 | Scope for [11]C-2,2,2-trifluoroethoxylation of arenes.** LG = leaving group. Yields are based on HPLC analyses of reaction mixtures. All yields are based on [11]C fluoroform conversion into the products, decay-corrected and expressed as mean ± SD (*n* = 3). Radioactive products were collected at least once for each substrate to check that HPLC yields matched isolated yields. **A** Substrate scope for heteroarenes using reaction condition (**A**). **B** Substrate scope for homoarenes using reaction condition (**B**). Auxiliary derived from Meldrum's acid.

Palbociclib ([11]**22**)], the herbicide Clopyralid ([11]**17**), and a purine derivative ([11]**16**)] in moderate to very good yields. Notably, [11]C-trifluoroethoxylation occurred preferentially at the more electron-deficient site (e.g., the *ortho-* vs *meta-*pyridinyl site for [11]**17**) or aryl ring (e.g., the pyridinyl *vs* homoarene ring in [11]**6** and [11]**20**) as expected for aromatic nucleophilic substitution reactions.

We found that homoarene precursors with common leaving groups were more challenging for [11]C-trifluoroethoxylation than the reactive heteroarenes[42]. We anticipated that the use of a more powerful hypervalent aryliodonium leaving group[43,44] could alleviate this issue. We opted to explore this possibility for the one-pot [11]C-trifluoroethoxylation of homoarene substrates. First, we screened

conditions for the reaction of [$^{11}$C]potassium 2,2,2-trifluoroethoxide with an iodonium ylide derived from Meldrum's acid (naphthalen-1-yl(2,4,6-trimethoxyphenyl)iodonium tosylate) (Supplementary Table 3). We found that treating a mixture of [$^{11}$C]CF$_3$CH$_2$OK with the iodonium salt precursor (55 μmol) in DMF at 60 °C for 3 min gave a high and optimal yield of the desired [$^{11}$C]**2** (87%; Supplementary Table 3, entry 2). The 2,4,6-trimethoxyphenyl group served as an effective aryl spectator ring; no [$^{11}$C]1,3,5-trimethoxy-(2-(2,2,2-tri-fluoroethoxy))benzene, was produced. An increase in temperature did not improve the yield of [$^{11}$C]**2**. Reduction in precursor amount reduced yield. Given the high yield obtained for [$^{11}$C]**2** with this approach under optimal conditions, we proceeded to explore substrate scope. Substituent electronics had substantial influence on reaction yields ([$^{11}$C]**23**–[$^{11}$C]**32**). Electron-withdrawing groups in *ortho*- and *para*-position gave high yields for the $^{11}$C-trifluoroethoxylation (Fig. 3B). $^{11}$C-Trifluorethoxylation yields were lower for substrates with *para*-electron-donating or *meta*-electron-withdrawing groups. Novel cross-coupling synthons, [$^{11}$C]**24**–[$^{11}$C]**27**, were obtained in useful yields. We draw attention to the syntheses of [$^{11}$C]**26** and [$^{11}$C]**28**, where mesityl was used as the partner aryl ring in the iodonium salt precursor[11].C-Trifluoroethoxylation was directed to the other aryl ring. This is an interesting observation because here the ring chemoselectivity is opposite to that seen for the non-copper mediated radio-fluorination of aryl(mesityl)iodonium salts[45].

We also explored aryliodonium ylides as precursors. $^{11}$C-Trifluoroethoxylation of three model ylides gave [$^{11}$C]**33**–[$^{11}$C]**35**, in moderate to high yields. Moreover, [$^{11}$C]**36**, an analog of the anti-diabetic drug empagliflozin (®Jardiance) was also obtained in moderate yield (47%) from an ylide. This exemplifies how iodonium ylides can serve as precursors for trifluoroethoxylation reactions. In addition, two activated fluoroarenes, with *ortho*-electron-deficient aryl rings, gave [$^{11}$C]**37** and [$^{11}$C]**38** in moderate to good yields where fluoride was the leaving group.

## $^{11}$C-2,2,2-Trifluoroethoxylation of aliphatic substrates

We were further interested in whether $^{11}$C-2,2,2-trifluoroethoxylation would occur on aliphatic substrates as well as arenes. This consideration prompted us to investigate the reactivity of $^{11}$CF$_3$CH$_2$OK with aliphatic substrates. We started with a model compound, a precursor to Posaconazole (®Noxafil) with a tosylate leaving group to optimize precursor amount and reaction temperature (Supplementary Table 4). $^{11}$C-Trifluoroethoxylation produced excellent yields of [$^{11}$C]**45** (89%) under conditions found to be optimal for hypervalent iodonium precursors (Supplementary Table 4, entry 3). Again, yield did not increase with temperature (Supplementary Table 4, entry 4). Reduction in precursor amount drastically diminished yield (Supplementary Table 4, entries 5 and 6). We next tested reaction scope by attempting to prepare a range of $^{11}$C-labeled alkyl-2,2,2-trifluoroethyl ethers from aliphatic precursors, including fourteen $^{11}$C-labeled biomolecules (Fig. 4). [$^{11}$C]**41** was obtained from a long chain iodoalkyl precursor in acceptable yield (45%). Benzyl halide and α-chloroacetyl precursors were readily converted into their analogous $^{11}$C-2,2,2-trifluoroethoxy ethers ([$^{11}$C]**42**, [$^{11}$C]**47**, [$^{11}$C]**48**, [$^{11}$C]**51**, [$^{11}$C]**52**) in high yields (53–89%) with good tolerance of other functional groups. Aliphatic tosylates derived from a variety of commercially available drug-like molecules MCPA (2-methyl-4-chlorophenoxyacetic acid), Helional, Ketoconazole, Bendazac, an α-D-glucopyranoside derivative, Oxaprozin, Ospemifene, Pterostilbene, and Cyhalofop-butyl reacted readily with $^{11}$CF$_3$CH$_2$OK to provide the desired $^{11}$C-2,2,2-trifluoroethoxy ethers, [$^{11}$C]**39**, [$^{11}$C]**40**, [$^{11}$C]**43**–[$^{11}$C]**46**, [$^{11}$C]**49**, [$^{11}$C]**50**, and [$^{11}$C]**53**–[$^{11}$C]**56**, in moderate to excellent yields (34–93%). Precursors with leaving groups attached to an ethylene glycol linker gave excellent yields of $^{11}$C-2,2,2-tri-fluoroethoxy ethers. Notably, aliphatic $^{11}$C-trifluoroethoxylation occurred in preference to reaction at aromatic sites.

## Determination of molar activity of [$^{11}$C]1 as a model compound

We measured the molar activity for a model product [$^{11}$C]**1**, produced by the $^{11}$C-trifluoroethoxylation of 2-chloropyrimidine, to verify that this labeling technique is no-carrier-added (NCA) and gives high molar activity. Starting with about 10 GBq of cyclotron-produced [$^{11}$C] methane that has been produced from a 10 μA × 10 min cyclotron irradiation, [$^{11}$C]**1** was obtained with a molar activity of 60 GBq/μmol, corrected to the end of radionuclide production (ERP). Such a high molar activity from a relatively limited cyclotron irradiation shows that the labeling reaction is invulnerable to carrier addition and dilution of molar activity[15,39].

## Synthesis of [$^{18}$F]potassium 2,2,2-trifluoroethoxide

Fluorine-18 labeling of PET tracers at aliphatic carbon by nucleophilic substitution of a good leaving group with [$^{18}$F]fluoride[19] can often lead to an [$^{18}$F]fluoroalkyl group that is vulnerable to radio-defluorination in vivo and to accumulation of [$^{18}$F]fluoride ion in the bone including skull. This can hamper accurate quantification of tracer uptake, especially in brain[20,46,47].$^{18}$F-Labeling in a 2,2,2-tri-fluoroethoxy group instead of an $^{18}$F-fluoroalkyl group could be a strategy to circumvent this issue. Having established an efficient route for $^{11}$C-trifluoroethoxylation, we next focused on translation of this labeling method from carbon-11 to fluorine-18 with a few representative substrates. For this purpose, [$^{18}$F]fluoroform was produced from no-carrier-added [$^{18}$F]fluoride and difluoroiodomethane[48]. CF$_2$$^{18}$FCH$_2$OK was generated by treatment of the [$^{18}$F]fluoroform with paraformaldehyde and *t*-BuOK in DMF in >95% yield. The reactivity of CF$_2$$^{18}$FCH$_2$OK was assessed under the optimal conditions found for $^{11}$C-trifluoroethoxylations (Fig. 5).$^{18}$F-Trifluoroethoxylations of aryl and aliphatic precursors proceeded smoothly and provided corresponding products in moderate to excellent yields similar to those from $^{11}$C-trifluoroethoxylation. Heteroarenes, such as pyridine, qui-noline, pyrimidine, isothiazole, and 1,3,5-triazine, with halogen leaving groups were converted into the corresponding $^{18}$F-2,2,2-trifluoroethoxy ethers in high yields (82–96%; Fig. 5A). Dependency of labeling position on aryl ring position of the leaving group (e.g., *ortho*- vs *meta*- as in [$^{18}$F]**4** and [$^{18}$F]**17**) or on the nature of the aryl ring (e.g., [$^{18}$F]**20**) was as seen for $^{11}$C-labeling. Heteroaryl rings with more structural complexity and diverse functionality were conveniently labeled at room temperature within 5 min and produced the corresponding $^{18}$F-labeled compounds in excellent yields ([$^{18}$F]**15**, [$^{18}$F]**17**–[$^{18}$F]**22**; 56–77%; Fig. 5A). These results indicate high potential for application of this labeling method to prospective structurally complex PET tracers. Homoarene precursors, including diaryliodonium salts ([$^{18}$F]**27** and [$^{18}$F]**31**), aryliodonium ylides ([$^{18}$F]**35** and [$^{18}$F]**36**), and fluoro precursors ([$^{18}$F]**37** and [$^{18}$F]**38**), afforded useful yields of $^{18}$F-labeled products (27–69%; Fig. 5B), as for the $^{11}$C-trifluoroethoxylations. Remarkably, the unprotected hydroxyl group in the Ataluren precursor was well tolerated ([$^{18}$F]**38**) showing compatibility of this labeling protocol to sensitive functionality.

Furthermore, we were keen to know whether potassium $^{18}$F-2,2,2-trifluoroethoxide could be useful for labeling at aliphatic carbon, given the limited availability of methods for constructing stable alkyl-CF$_2$$^{18}$F bonds[49]. In this regard, we tested identical reaction conditions to those used for $^{11}$C-trifluoroethoxylation on diverse aliphatic substrates, prepared from drugs, herbicides, and other biomolecules. To our delight, this protocol successfully enabled the installation of a $^{18}$F-2,2,2-tri-fluoroethoxy moiety onto aliphatic carbon in a variety of complex structures in acceptable to excellent yields ([$^{18}$F]**44**–[$^{18}$F]**46**, [$^{18}$F]**49**–[$^{18}$F]**56**; 15–95%; Fig. 5C). Taken together, these results show that this methodology, based on the transformation of fluoroform into potassium 2,2,2-trifluoroethoxide and subsequent functionalization of aliphatic carbons, is equally versatile for both carbon-11 and fluorine-18 with the same non-radioactive precursor.

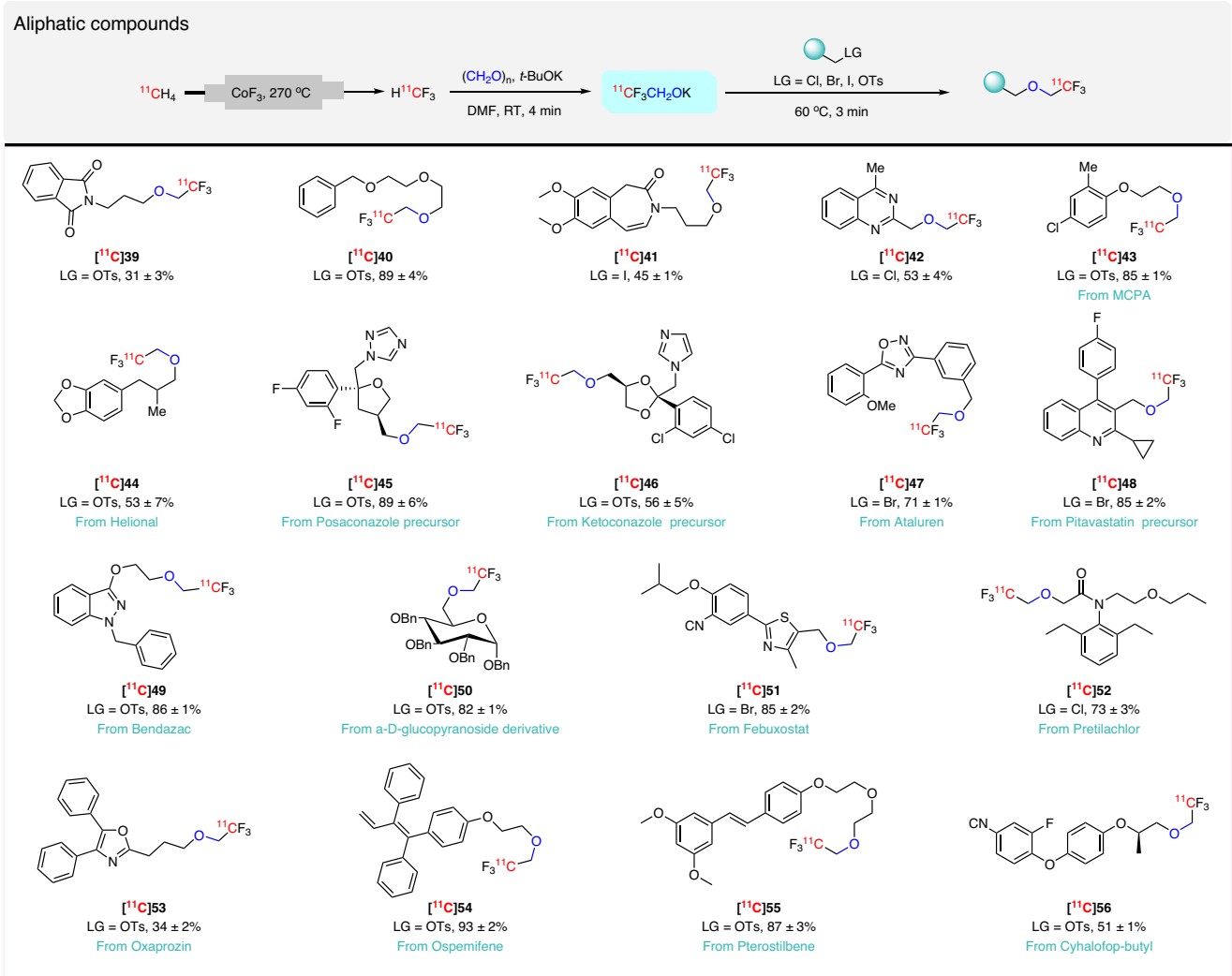

**Fig. 4 | Scope for ¹¹C-2,2,2-trifluoroethoxylation of aliphatic substrates.** LG = leaving group, Decay-corrected yields are based on HPLC analyses of crude reaction mixtures. All yields are based on [¹¹C]fluoroform conversion into the products and expressed as mean ± SD ($n = 3$). Radioactive products were collected at least once for each substrate to check that HPLC yields matched isolated yields.

## Determination of molar activity of [¹⁸F]1 as a model compound

We measured the molar activity of [¹⁸F]**1** to be 1.3 GBq/μmol, decay-corrected. The method of [¹⁸F]fluoroform synthesis that we used was one reported in the literature[48] and known to give low molar activity. We did not observe any significant release of fluoride ion in the production of [¹⁸F]potassium 2,2,2-trifluoroethoxide under basic conditions, as evidenced by absence of [¹⁸F]fluoride at the solvent front in the HPLC analysis of derived [¹⁸F]2,2,2-trifluoroethanol (Supplementary Fig. 17). Therefore, the molar activity of the starting [¹¹C]fluoroform determines the molar activity of ¹⁸F-labeled 2,2,2-trifluoroethoxy products.

## Synthesis of isotopologues of potassium 2,2,2-trifluoroethoxide

Isotopologues differ only in their isotopic substitutions and play an important role in drug development[50]. Deuteration[51] is widely practiced to improve the metabolic stability of PET tracers in ¹⁸F-fluoroalkyl positions. ¹³C-Labeling enables investigations of drug pharmacokinetics and metabolism by ¹³C-NMR spectroscopy and mass spectrometry[52,53]. Simple methods for accessing stable isotopically labeled compounds are highly desirable. The availability of isotopically labeled fluoroforms (H¹³CF₃, H¹¹CF₃, and HCF₂¹⁸F) and paraformaldehyde [(CD₂O)$_n$ and (¹³CH₂O)$_n$] and our method for the in situ generation of CF₃CH₂OK from fluoroform, provide an opportunity to explore the incorporation of isotopically labeled 2,2,2-trifluoroethoxy groups into a diverse array of substrates. For demonstration, we synthesized isotopologues of **57**, an analog of a well-known COX-1 PET tracer, [¹¹C] PS13[32]. Compounds [¹¹C]**57** and [¹⁸F]**57** were readily obtained by treating a tosylate precursor with [¹¹C/¹⁸F]CF₃CH₂OK under optimized conditions. The reaction was equally effective when substituting paraformaldehyde with (CD₂O)$_n$ and (¹³CH₂O)$_n$, leading to high yield syntheses of [²H]**57**, [²H/¹¹C]**57**, [²H/¹⁸F]**57**, [¹³C]**57**, [¹³C/¹¹C]**57**, and [¹³C/¹⁸F]**57** (Fig. 6). Hence, this isotope labeling protocol has exceptional potential for broad application.

In summary, based on the transformation of paraformaldehyde with fluoroform, we devised a highly effective one-pot method for appending a wide range of aryl, heteroaryl, and aliphatic organic compounds with an isotopically labeled 2,2,2-trifluoroethoxy group. Especially, reaction of paraformaldehyde with [¹¹C]fluoroform or [¹⁸F] fluoroform efficiently provides ¹¹CF₃CH₂OK and ¹⁸FF₂CCH₂OK, respectively, as broadly useful no-carrier-added labeling synthons with ability to produce candidate PET tracers bearing either a ¹¹C- or ¹⁸F-labeled 2,2,2-trifluoroethoxy group. Use of paraformaldehyde and fluoroform labeled with stable isotopes (²H, or ¹³C) gives ready access to isotopologues of 2,2,2-trifluoroethoxy compounds. Consequently, the 2,2,2-trifluoroethoxy group may garner increasing interest for both drug and PET tracer development.

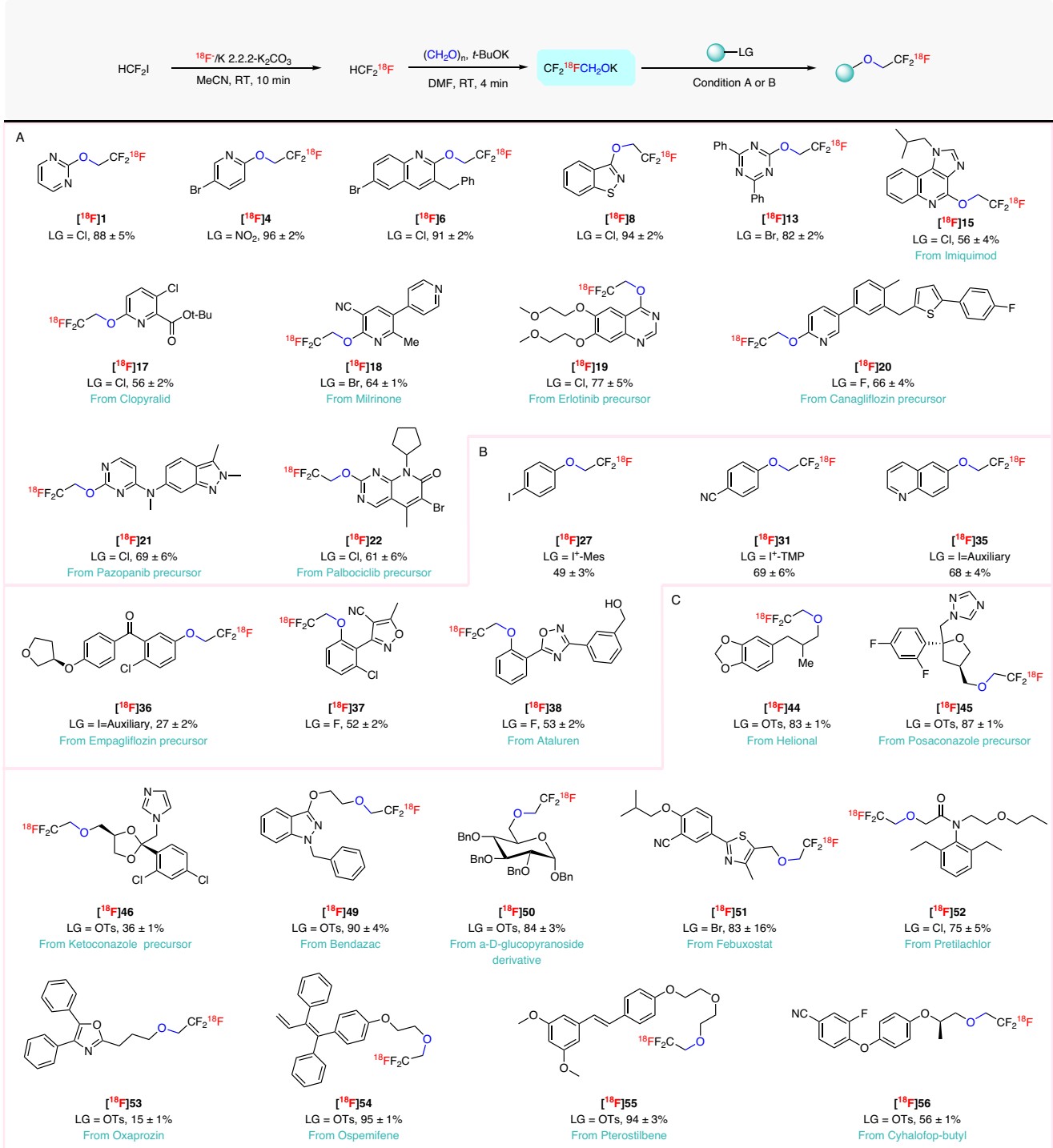

**Fig. 5 | Scope for the 18F-2,2,2-trifluoroethoxylation of aromatic and aliphatic substrates.** LG = leaving group. Yields are calculated from HPLC analyses of crude reaction mixtures and based are [18F]fluoroform conversion into the products. Radioactive products were collected at least once for each substrate to confirm that HPLC yields match isolated yields. All yields are decay-corrected and reported as mean ± SD for n = 3. **A** Substrate scope for homoarenes using reaction condition (**B**). **B** Substrate scope for heteroarenes using reaction condition (**B**). **C** Substrate scope for aliphatic compounds using reaction condition (**B**).

## Methods

### General procedure for the one-pot 2,2,2-trifluoroethoxylation of aromatic and aliphatic precursors from fluoroform and isotopically labeled paraformaldehyde

Isotopically labeled ($^{13}$C and $^2$H) paraformaldehyde (3.0 equiv.) and t-BuOK (3.0 equiv.) were added to a round-bottomed flask equipped with a magnetic stirrer bar followed by DMF (6 mL/ 1 mmol) under argon atmosphere. A solution of fluoroform (1.0 equiv., 0.3 M) in DMF was added and the reaction mixture was stirred at RT for 3 h. Precursor solution in DMF was added, and the reaction mixture was stirred at RT or 60 °C for additional 2 h. The reaction was then quenched with water (2 mL) and the mixture was extracted with DCM (2 × 10 mL). The organic phase was washed with brine, dried (Mg$_2$SO$_4$), and concentrated under

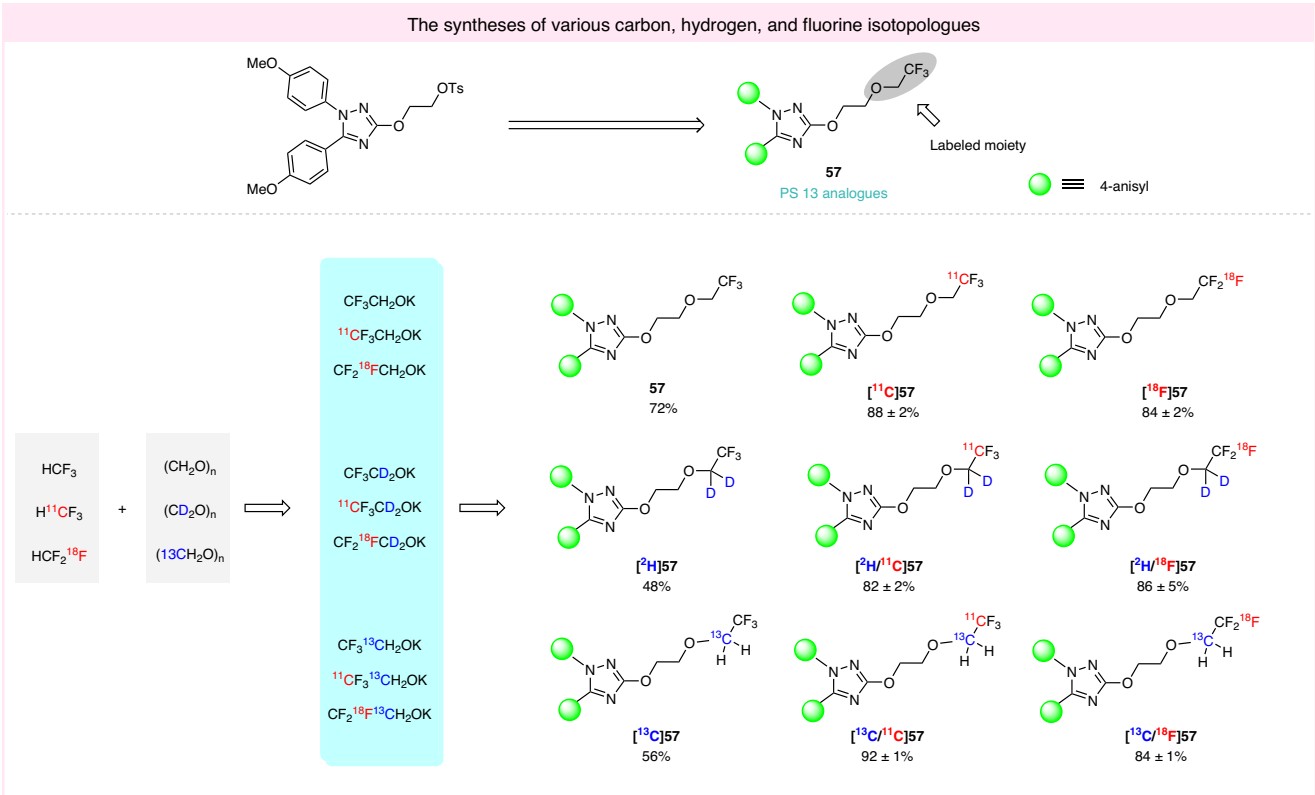

**Fig. 6 | The syntheses of nine isotopologues of a model compound (an analog of the COX-1 PET tracer [¹¹C]PS13).** Yields were based on HPLC analyses of crude reaction mixtures. Radioactive products were collected at least once with identity confirmed with LC-MS. HPLC radiochemical yields (RCYs) matched isolated RCYs and are reported as mean ± SD for $n = 3$.

reduced pressure. The residue was purified by flash chromatography on silica gel to give the 2,2,2-trifluoroethyl ether. Purification methods and characterization data of all the individual trifluoroethoxy compounds synthesized in this work can be obtained in the Supplementary Methods Section 1.2, 1.3 and 2.4.

**Synthesis of [¹¹C]Fluoroform[22]**

[¹¹C]Methane (3.7–5.5 GBq) was produced from a cyclotron (PET-trace; GE Healthcare) according to the ¹⁴N(p.α)¹¹C reaction by irradiating nitrogen (130 psi) containing hydrogen (7%) and then trapped in a U-tube packed with Porapak Q (1 g, 80–100 mesh) that was being cooled with liquid argon. Any untrapped radioactivity was captured in a waste bag for safety. When radioactivity in the cooled Porapak Q trap had maximized, as indicated by a nearby radiation detector, the liquid argon coolant was removed. The trap was then allowed to warm to room temperature while being purged with a controlled flow of helium (20 mL/min) to direct and pass the [¹¹C]methane sequentially through a Sicapent column (to remove any moisture), a CoF₃ column heated at 270 °C, a cooled trap for HF, and finally into a [¹¹C]fluoroform collection trap being cooled in liquid argon. This transfer generally took about 15 min. Then the [¹¹C]fluoroform trap was placed in a warm water bath (60 °C) for another 35–45 s to release [¹¹C] fluoroform into a vial containing DMF (0.8 mL) at –40 °C. The yield of [¹¹C]fluoroform from [¹¹C]methane was usually 35–55% accompanied by 10–50% [¹¹C]fluoromethane (Supplementary Fig. 12). This [¹¹C]fluoroform solution in DMF was used for further reactions. The percentage of radioactivity represented by [¹¹C] fluoroform in the HPLC analyte was calculated from the radio-HPLC chromatogram from peak areas with a correction for radioactive decay between radioactive peaks during the analysis.

The yields of the reaction products are based on the conversion of [¹¹C]fluoroform into radioactive products and are calculated by decay-correction to the beginning of the HPLC analysis. We confirmed that all the radioactivity injected onto the HPLC column was fully recovered.

**Synthesis of [¹⁸F]Fluoroform[48]**

[¹⁸F]Fluoride ion was produced on a cyclotron (PETtrace; GE Healthcare) according to the ¹⁸O(p,n)¹⁸F reaction by irradiating ¹⁸O-enriched water (3 mL, 98 atom%) with a beam of protons (16.5 MeV; 35–45 μA) for at least 75 min. [¹⁸F]Fluoroform was synthesized within a lead-shielded hot-cell with a fully automated apparatus (TRACERlab™ FX2N; GE Healthcare). Thus, [¹⁸F]fluoride ion (3.72–11.1 GBq) in [¹⁸O]water (200–400 μL) and a solution (100 μL) containing K₂CO₃ (3.4 μmol) plus K 2.2.2 (13.6 μmol) were loaded into a glass vial. MeCN (2 mL) was added, and the solvent was azeotropically removed at 80–100 °C under a stream of nitrogen gas that was vented to vacuum. This step was repeated after a second addition of MeCN (2 mL). A solution of difluoroiodomethane (8.0 mg, 45 μmol) in anhydrous acetonitrile (1.0 mL) was then added to the dried [¹⁸F]fluoride-K₂CO₃/K 2.2.2 complex, sealed, and heated at 35 °C for 10 min. [¹⁸F]Fluoroform was flushed out of the vial with helium (20 mL/min) and into the [¹⁸F]fluoroform trap. The transfer generally required 5 min. Then the [¹⁸F]fluoroform trap was put in warm water bath (60 °C) for another 35–45 s to release [¹⁸F]fluoroform into a vial containing DMF (0.8 mL). This DMF solution of [¹⁸F]fluoroform was used for subsequent reactions. The yield of [¹⁸F]fluoroform produced by this method generally ranged between 35–65% from dried [¹⁸F]fluoride with >98% radiochemical purity (Supplementary Fig. 15).

### General procedure for the [11]C- and [18]F- 2,2,2-trifluoroethoxylation of aromatic and aliphatic precursors from isotopically labeled fluoroform and paraformaldehyde

About 30 min before the end of radionuclide production, a mixture of t-BuOK (112.2 mg, 1.0 mmol) and paraformaldehyde (15.0 mg, 0.5 mmol) in DMF (6 mL) was prepared in a glovebox under argon and kept there until close to the start of radiochemistry. The t-BuOK-paraformaldehyde reagent mixture in DMF (200 µL) was added to a 1-mL V vial, septum-sealed, removed from the glovebox, and transferred to the lead-shielded hot-cell for the radiochemistry. [11C]Fluoroform or [18F]fluoroform (37–296 MBq) in DMF (50–300 µL) was added to the vial, mixed, and left at RT for 4 min. A DMF solution (200 µL) of the aromatic or aliphatic precursor (55 µmol) was then added. The reaction mixture was kept at RT for 1 min or heated at 60 °C for 3 min and then quenched with water (100 µL). This crude product was analyzed with radio-HPLC as detailed in Supplementary Methods section. Areas of all the radiochemical product peaks were decay-corrected to the beginning of the HPLC analysis for radiochemical yields calculations. HPLC analysis of each [11]C- and [18]F- labeled trifluoroethoxylated products and the respective HPLC chromatograms can be obtained in Supplementary Methods Sections 6 and 7.

### Data availability

Details about materials and methods, experimental procedures including organic syntheses and radiochemistry, and NMR spectra and HPLC chromatographs are available in the Supplementary Information. Any further queries on the data can be directed to either S.T or V.W.P.

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

## Acknowledgements
We acknowledge the Intramural Research Program of the National Institutes of Health (NIMH; ZIA-MH002793) for financial support and thank the NIH Clinical Center (Chief Dr. P. Herscovitch) for radioisotope production.

## Author contributions
Q.Z., S.T., S.J., C.L.M., contributed to study design and experimental implementation. S.T. and V.W.P. contributed to study design and supervision. All authors contribute to writing and review of the manuscript.

## Competing interests
The authors declare no competing interests.
