## [Peer Review File · Nature Communications]

Isotopologues of Potassium 2,2,2-Trifluoroethoxide for Applications in Positron Emission Tomography and BeyondREVIEWER COMMENTS

Reviewer #1 (Remarks to the Author):

Dr. Pike and coworkers developed a mild, efficient, and versatile radiolabeling methodology for one-pot, two-stage 2,2,2-trifluoroethylations of aromatic and aliphatic precursors. These methods covered a broad scope, including leaving groups, the complexity of molecules, and isotopes. Various drug-like molecules have also been successfully radiolabeled with either C11 or F18. Furthermore, stable isotopes have been involved in producing potassium 2,2,2-trifluoro ethoxide. The results in this manuscript are comprehensive and demonstrate the potential application in efficient radiopharmaceutical assembling and drug discovery. Overall, the manuscript was well-written, and the data in Si are well-collected and organized. From my point of view, the operation procedures and devices used in this research might be challenging even for experienced radiochemists. I would recommend that the submitted manuscript is acceptable in publication after major revision.

Remarks to the author

Page 8, line 140/Page 8, line 155/ Page 8, line 158

e.g. "Notably, 11C-trifluoroethoxylation occurred preferentially at the more electron-deficient site (e.g., the ortho vs meta pyridinyl site for [11C]17) or aryl ring (e.g., the pyridinyl vs homoarene ring in [11C]6 and [11C]20)."

In several reactions, the author only showed the results or described the selectivity in the position of arene without further detailed explanation about the mechanism or reason.

Page 9, line 173

"18F-Labeling in a 2,2,2-172 trifluoroethoxy group instead of an 18F-fluoroalkyl group could be a strategy to circumvent this issue. This consideration prompted us to investigate the reactivity of 11CF₃CH₂OK with aliphatic substrates."

This gave the reader a misunderstanding that you will start to introduce the 18F methodology.

The author mentioned in the manuscript that "Radioactive products were collected at least once for each substrate to check that HPLC yields matched isolated yields." Is there any mismatch between the HPLC yield and the isolated yield? I have not seen any data in SI. In my experience, the isolated yield is usually lower than the HPLC yield. Also, the author does not clearly describe the method for radiochemical yield determination. It is difficult to determine the ratio between [11C]fluoroform and [11C]fluoromethane from the spectrum (very broad peak for fluoromethane) and accurate radio activities. The yield determined by [11C]fluoroform conversion from the HPLC spectrum is more appropriate for "RCC".

In the analytical HPLC, the spectrum showed that some standard compounds were not pure. e.g. 10 (S69), 11, 12, 15, 23, 25, 38, 39, 41, 44, 46, 49, 50, 51, 54, etc.

The coinjection analytical HPLC spectrums are missing for compounds 23 (S82), 1(s116).

Page 5, line 107

"For this purpose, we routinely [11C]fluoroform by CoF₃-mediated fluorination of cyclotron-produced [11C]methane."

Verb missing

Reviewer #2 (Remarks to the Author):

In this article, Telu, Pike and coworkers report the preparation of potassium 2,2,2-trifluoroethoxide by reaction of fluoroform with paraformaldehyde and potassium tert-butoxide. The potassium 2,2,2-trifluoroethoxide thus obtained can be directly used as a trifluoroethoxylation reagent and convert a wide range of substrates into trifluoroethoxylated compounds via nucleophilic substitution reactions. Furthermore, the strategy can be applied to the preparation of a number of organic compounds bearing an isotopically labeled 2,2,2-trifluoroethoxy group by use of [^{11}C]fluoroform or [^{18}F]fluoroform.

The conversion of fluoroform to potassium 2,2,2-trifluoroethoxide is not surprising given the literature reports on the reaction of fluoroform with aldehydes. Practically, it is not useful because 2,2,2-trifluoroethanol is pretty cheap and easily available. The subsequent nucleophilic substitution reactions using $\text{CF}_3\text{CH}_2\text{OK}$ as the nucleophile are also well expected. However, the ability to incorporate an isotope, either C-11 or F-18, into the trifluoroethoxy group demonstrates the uniqueness and potentials of the method. And I expect that this work will be of interest to a wide audience.

The isotopically labeled compounds reported in this manuscript are well characterized. The manuscript is well written. Therefore, I'd like to recommend this work to be published in Nature Communications.

One minor point: The authors describe their reaction as "...metal-free conversion of fluoroform with paraformaldehyde into highly reactive potassium 2,2,2-trifluoroethoxide". The POTASSIUM is there and you still claim it is metal-free?

Reviewer #3 (Remarks to the Author):

In this manuscript, Pike, Telu and co-workers we described a conversion of fluoroform with paraformaldehyde into highly reactive potassium 2,2,2- ^{12}C trifluoroethoxide ($\text{CF}_3\text{CH}_2\text{OK}$) and then the applications of this fluorinated salt for 2,2,2-trifluoroethoxylations of both aromatic and aliphatic precursors. In one-pot, two-stage were developed. Furthermore, this protocol was used for labeling fluoroform with either carbon-11 ($t_{1/2} = 15.204$ min) or fluorine-18 ($t_{1/2} = 109.8$ min). Finally, the reaction of paraformaldehyde with [^{11}C]fluoroform or [^{18}F]fluoroform efficiently provides $^{11}\text{CF}_3\text{CH}_2\text{OK}$ and $^{18}\text{FF}_2\text{CH}_2\text{OK}$, respectively, as new, and broadly useful no-carrier-added labeling synthons, with ability to produce novel PET tracers bearing either a ^{11}C - or ^{18}F -labeled 2,2,2-trifluoroethoxy group. This work was interesting, and this manuscript was organized and written well. Of course, this manuscript was recommended for publication in Nature Communication when the following comments were made.

(1) Cobalt trifluoride should be used in the fluorination reaction of $^{11}\text{CH}_4$. Please correct the molecular formula.

(2) The reaction shown in Fig. 1 (A) is not closely relevant to the main topic of this manuscript. Considering that the cold reaction can be more conveniently achieved through trifluoroethanol, it is suggested to change it to a comparison between the labeling methods of ^{11}C and ^{18}F in this manuscript and previous methods to demonstrate the novelty.

(3) The specific activity data (molar activity) was included in the ^{11}C -related reaction, but not shown in ^{18}F -related reactions. It would be better to provide these data. What is the stability

of the ^{18}F -labeled trifluoroethoxy anion during the reaction? Is there any exchange between ^{18}F and ^{19}F , which leads to the decrease of specific activity? Do some substrates promote the decomposition of ^{18}F -labeled trifluoroethoxy anion, resulting in a decrease in molar activity? It is recommended to give some explanations.

(4) What is the purpose of synthesizing the deuterated trifluoroethoxy molecules? Is it because the metabolic stability of ^{18}F -labeled trifluoroethoxy compounds is not good enough and prone to defluorination? It would be better to give some explanations.

(5) Besides paraformaldehyde, are these methods developed in this manuscript also applicable to other aldehydes?

(6) What is the antimetabolic stability of the molecules with potential biological activity shown in this paper? It is suggested to evaluate the possibility of defluorination in vivo by PET of small animals.

We thank the Reviewers for thorough review of our manuscript on the “*Isotopologues of Potassium 2,2,2-Trifluoroethoxide for Applications in Positron Emission Tomography and Beyond*”.

We appreciate your positive review and the comments to improve the manuscript.

Here is our response to the reviewers and editor’s comments and how we addressed them. Matters needing attention by reviewers r are highlighted in yellow; our responses to reviewers r are in blue type and new material or changes incorporated are in red type.

Response to the reviewer’s comments

Reviewer: 1

Comments:

Dr. Pike and coworkers developed a mild, efficient, and versatile radiolabeling methodology for one-pot, two-stage 2,2,2-trifluoroethoxylations of aromatic and aliphatic precursors. These methods covered a broad scope, including leaving groups, the complexity of molecules, and isotopes. Various drug-like molecules have also been successfully radiolabeled with either C11 or F18. Furthermore, stable isotopes have been involved in producing potassium 2,2,2-trifluoro ethoxide. The results in this manuscript are comprehensive and demonstrate the potential application in efficient radiopharmaceutical assembling and drug discovery. Overall, the manuscript was well-written, and the data in Si are well-collected and organized. From my point of view, the operation procedures and devices used in this research might be challenging even for experienced radiochemists. I would recommend that the submitted manuscript is acceptable in publication after major revision.

We thank the reviewer for the positive comments on the manuscript.

Remarks to the author

- Page 8, line 140/Page 8, line 155/ Page 8, line 158

e.g. “Notably, ¹¹C-trifluoroethoxylation occurred preferentially at the more electron-deficient site (e.g., the *ortho* vs *meta* pyridinyl site for [¹¹C]17) or aryl ring (e.g., the pyridinyl vs homoarene ring in [¹¹C]6 and [¹¹C]20).”

In several reactions, the author only showed the results or described the selectivity in the position of arene without further detailed explanation about the mechanism or reason.

We have changed the text in the results and discussion part on the ¹¹C-trifluoroethoxylation of aromatic substrates to provide a more detailed explanation on reaction selectivity. The selectivity for arene ¹¹C-trifluoroethoxylation is as expected for aromatic nucleophilic substitution. When two possible leaving groups are the same, trifluoroethoxylation occurs at the most electron-deficient ring (e, g., as in [¹¹C]20) or the most activated ring position (e.g. *ortho* vs *meta* in pyridinyl group, as in, [¹¹C]17). If the leaving groups are different, trifluoroethoxylation preferentially occurs in accord with the leaving group ability in S_NAr reactions (NO₂ > Br; e.g. [¹¹C]4) or most activated ring (e.g., [¹¹C]6) or ring position. The text in the manuscript at page 8 is revised to now read:

Notably, ¹¹C-trifluoroethoxylation occurred preferentially at the more electron-deficient site (e.g., the *ortho* vs *meta* pyridinyl site for [¹¹C]17) or aryl ring (e.g., the pyridinyl vs homoarene ring in [¹¹C]6 and [¹¹C]20), as expected for aromatic nucleophilic substitution reactions.

- Page 9, line 173

“¹⁸F-labeling in a 2,2,2-trifluoroethoxy group instead of an ¹⁸F-fluoroalkyl group could be a strategy to circumvent this issue. This consideration prompted us to investigate the reactivity of ¹¹C-F₃CH₂OK with aliphatic substrates.”

This gave the reader a misunderstanding that you will start to introduce the ¹⁸F methodology.

We agree with the reviewer and have now changed the description. Our intention was to indicate that radiolabeled trifluoroethoxy groups are metabolically more stable than the radiolabeled fluoroethoxy groups. Because the trifluoroethoxy moiety could be labeled with either C-11 and F-18, we used this analogy comparing the metabolic stability of trifluoroethoxy group with that of fluoroethoxy group (or fluoralkoxy moiety). We now removed the part where we described F-18 labeling in page 9 to the page 11 under the subheading “Synthesis of [¹⁸F]potassium 2,2,2-trifluoroethoxide” to eliminate any possible misunderstanding. The following changes were made to the text in response to the reviewer’s comments: Page 9:

¹¹C-2,2,2-trifluoroethoxylation of aliphatic substrates

We added “We were further interested in whether ¹¹C-2,2,2-trifluoroethoxylation would occur an aliphatic substrate as well as arenes.”

Page 11, the following material is mainly moved from page 9 – to separated discussion of ¹¹C and ¹⁸F:

Synthesis of [¹⁸F]potassium 2,2,2-trifluoroethoxide

Fluorine-18 labeling of PET tracers at aliphatic carbon by nucleophilic substitution of a good leaving group with [¹⁸F]fluoride (19) can often lead to an [¹⁸F]fluoroalkyl group that is vulnerable to radiodefluorination *in vivo* and to accumulation of [¹⁸F]fluoride ion in the bone including skull. This can hamper accurate quantification of tracer uptake, especially in brain (20,46,47). ¹⁸F-labeling in a 2,2,2-trifluoroethoxy group instead of an ¹⁸F-fluoroalkyl group could be a strategy to circumvent this issue.

- The author mentioned in the manuscript that “Radioactive products were collected at least once for each substrate to check that HPLC yields matched isolated yields.” Is there any mismatch between the HPLC yield and the isolated yield? I have not seen any data in SI. In my experience, the isolated yield is usually lower than the HPLC yield. Also, the author does not clearly describe the method for radiochemical yield determination. It is difficult to determine the ratio between [¹¹C]fluoroform and [¹¹C]fluoromethane from the spectrum (very broad peak for fluoromethane) and accurate radio activities. The yield determined by [¹¹C]fluoroform conversion from the HPLC spectrum is more appropriate for “RCC”.

We have collected radioactive products at least once for each substrate to measure isolated yield. Generally, the isolated yields are slightly lower than the HPLC yields but only by about 5–10% (as in the reviewer’s experience). This difference is likely due to peak collection errors.

We now indicated the isolated yields of each substrate in the revised SI.

Radiochemical yields were determined in the manner suggested by the reviewer, i.e. [¹¹C]fluoroform conversion to radioactive products from the HPLC chromatogram. The areas under each radio-product peak were decay-corrected to the start of the chromatogram for the calculation of the radiochemical yield. We have modified Figure 3, 4, 5, and 6 legends to clarify that yields are based on [¹¹C]fluoroform conversion to the products. For example, Figure 3 legend now reads:

All yields are based on ^{11}C fluoroform conversion to the products, decay-corrected and expressed as mean \pm SD ($n = 3$).

- In the analytical HPLC, the spectrum showed that some standard compounds were not pure. e.g. **10 (S69), 11, 12, 15, 23, 25, 38, 39, 41, 44, 46, 49, 50, 51, 54, etc.**

The non-radioactive impurities found in some of the analytes under co-injections are not the impurities from the reference standards. These non-radioactive impurities were co-eluted with the collected radioactive products from the HPLC purification of the crude reaction mixtures under non-optimized conditions for fast analyses of C-11 labeled products. We have now added HPLC chromatograms to the revised SI for all the reference compounds to allow comparison of their retention times with those of the radiolabeled products. More specifically, we added:

HPLC chromatograms for compounds **10, 11, 12, 15, 23, 25, 38, 39, 41, 44, 46, 49, 50, 51, and 54** to compare with analyses of ^{11}C **10 (S69), ^{11}C 11, ^{11}C 12, ^{11}C 15, ^{11}C 23, ^{11}C 25, ^{11}C 38, ^{11}C 39, ^{11}C 41, ^{11}C 44, ^{11}C 46, ^{11}C 49, ^{11}C 50, ^{11}C 51, ^{11}C 54, and also all the compounds we radiolabeled to the SI.”**

- The coinjection analytical HPLC spectrums are missing for compounds **23 (S82), 1(s116).**

We co-injected the sample from ^{11}C **1** labeling with **1**. For ^{18}F **1**, we compared its retention time with that of ^{11}C **1** and reference **1**. The HPLC retention time of ^{18}F **1** is the same as ^{11}C **1**. We also constructed a 5-point calibration curve for compound **1** for molar activity measurements on ^{11}C **1** and ^{18}F **1**. In all instances the retention times are same. We did not think it is necessary to co-inject the isolated ^{18}F **1** with reference standard to confirm its identity. In the case of ^{11}C **23**, the yield was so low ($12 \pm 2\%$) that we were not able to re-analyze with added reference standard **23**.

- Page 5, line 107

“For this purpose, we routinely ^{11}C fluoroform by CoF_3 -mediated fluorination of cyclotron-produced ^{11}C methane.” Verb missing

Thank you for finding this error. We have now corrected it. We added the verb “produced” to make complete meaningful sentence. The text (page 5) now reads:

For this purpose, we routinely produced ^{11}C fluoroform by CoF_3 -mediated fluorination of cyclotron-produced ^{11}C methane.

Reviewer:2 (Remarks to the Author):

In this article, Telu, Pike and coworkers report the preparation of potassium 2,2,2-trifluoroethoxide by reaction of fluoroform with paraformaldehyde and potassium tert-butoxide. The potassium 2,2,2-trifluoroethoxide thus obtained can be directly used as a trifluoroethoxylation reagent and convert a wide range of substrates into trifluoroethoxylated compounds via nucleophilic substitution reactions. Furthermore, the strategy can be applied to the preparation of a number of organic compounds bearing an isotopically labeled 2,2,2-trifluoroethoxy group by use of ^{11}C fluoroform or ^{18}F fluoroform. The conversion of fluoroform to potassium 2,2,2-trifluoroethoxide is not surprising given the literature reports on the reaction of fluoroform with aldehydes. Practically, it is not useful because 2,2,2-trifluoroethanol is pretty cheap and easily available. The subsequent nucleophilic substitution reactions using $\text{CF}_3\text{CH}_2\text{OK}$ as the nucleophile are also well expected. However, the ability to incorporate an isotope, either C-11 or F-18, into the trifluoroethoxy group demonstrates the uniqueness and potentials of the method. And I expect that this work will be of interest to a wide audience.

The isotopically labeled compounds reported in this manuscript are well characterized. The manuscript is well written. Therefore, I'd like to recommend this work to be published in Nature Communications.

The authors thank reviewer 2 for positive comments on the research presented in this manuscript. We appreciate your recommendation to publish this work in Nature Communications.

- One minor point: The authors describe their reaction as "...metal-free conversion of fluoroform with paraformaldehyde into highly reactive potassium 2,2,2-trifluoroethoxide". The POTASSIUM is there and you still claim it is metal-free?

Thank you for the suggestion. We have now replaced metal-free to transition metal-free conversion in three places in the manuscript.

Page 1 (Abstract) now reads:

Herein, we describe a novel, rapid, and transition metal-free conversion of fluoroform with paraformaldehyde into a highly reactive potassium 2,2,2-trifluoroethoxide....

at page 3:

Herein, we describe a novel, rapid, and transition metal-free conversion of fluoroform into highly reactive potassium 2,2,2-trifluoroethoxide

And at page 5:

We anticipate that this new transition metal-free transformation can find wide application in fluorine chemistry.

Reviewer #3 (Remarks to the Author):

In this manuscript, Pike, Telu and co-workers we described a conversion of fluoroform with paraformaldehyde into highly reactive potassium 2,2,2-trifluoroethoxide ($\text{CF}_3\text{CH}_2\text{OK}$) and then the applications of this fluorinated salt for 2,2,2-trifluoroethylations of both aromatic and aliphatic precursors. In one-pot, two-stage were developed. Furthermore, this protocol was used for labeling fluoroform with either carbon-11 ($t_{1/2} = 15.204$ min) or fluorine-18 ($t_{1/2} = 109.8$ min). Finally, the reaction of paraformaldehyde with [^{11}C]fluoroform or [^{18}F]fluoroform efficiently provides $^{11}\text{CF}_3\text{CH}_2\text{OK}$ and $^{18}\text{FF}_2\text{CCH}_2\text{OK}$, respectively, as new, and broadly useful no-carrier-added labeling synthons, with ability to produce novel PET tracers bearing either a ^{11}C - or ^{18}F -labeled 2,2,2-trifluoroethoxy group. This work was interesting, and this manuscript was organized and written well. Of course, this manuscript was recommended for publication in Nature Communication when the following comments were made.

The authors thank reviewer 3 for positive comments and critique on the research presented in this manuscript.

Cobalt trifluoride should be used in the fluorination reaction of $^{11}\text{CH}_4$. Please correct the molecular formula.

Thank you for finding this error. We now corrected the formula in the schemes above the Figure 3 and Figure 4 in the manuscript.

The formula of cobalt trifluoride in the schemes above Figure 3 and Figure 4 now reads " CoF_3 ."

The reaction shown in Fig. 1 (A) is not closely relevant to the main topic of this manuscript. Considering that the cold reaction can be more conveniently achieved through trifluoroethanol, it is suggested to change it to a comparison between the labeling methods of ^{11}C and ^{18}F in this manuscript and previous methods to demonstrate the novelty.

The authors respectfully decline to change Figure 1A. Figure 1A compares the literature transformations of fluoroform to the novel transformation of fluoroform in this work. Figure 1A reflects the main topic or theme of the work in the manuscript i.e., a novel transformation of fluoroform into a reactive synthon and its applications to trifluoroethoxylations, which we eventually translated to a radiochemistry platform as well as to stable isotopologues.

The specific activity data (molar activity) was included in the ^{11}C -related reaction, but not shown in ^{18}F -related reactions. It would be better to provide these data. What is the stability of the CF_2^{18}F anion during the reaction? Is there any exchange between ^{18}F and ^{19}F , which leads to the decrease of specific activity? Do some substrates promote the decomposition of CF_2^{18}F anion, resulting in a decrease in molar activity? It is recommended to give some explanations.

We now provide a molar activity measurement for ^{18}F 1. The molar activity is 1.3 GBq/ μmol is low but as we might expect from the literature method that we used to produce ^{18}F fluoroform (van der Born et al. *Ang. Chem. int. Ed*, 53, 11046-11050, 2014).

We have now added the molar activity determination of ^{18}F 1 to the SI in section 6.6."

In over 30 productions, HPLC analyses of unpurified ^{18}F CF₃CH₂OH (Figure S18) did not show a radioactivity peak at the solvent front for ^{18}F fluoride, showing absence of any ^{19}F fluoride release during the procedure. We add explanation and clarification to the manuscript (page 12), as follows:

Determination of molar activity of ^{18}F 1 as a model compound

We measured the molar activity of ^{18}F 1 to be 1.3 GBq/ μmol , decay corrected. The method of ^{18}F fluoroform synthesis that we used was one reported in the literature (48) and known to give molar activity of this order. We did not observe any significant release of fluoride ion in the production of ^{18}F potassium 2,2,2-trifluoroethoxide under basic conditions, as evidenced by absence of ^{18}F fluoride at the solvent front in the HPLC analysis of derived ^{18}F 2,2,2-trifluoroethanol (Figure S18). Therefore, the molar activity of the starting ^{18}F fluoroform determines the molar activity of ^{18}F -labeled 2,2,2-trifluoroethoxy products.

- What is the purpose of synthesizing the deuterated trifluoroethoxy molecules? Is it because the metabolic stability of ^{18}F -labeled trifluoroethoxy compounds is not good enough and prone to defluorination? It would be better to give some explanations.

The deuterated trifluoroethoxy compounds were synthesized to exemplify the method's broader applicability to stable isotopes (i.e., beyond radiolabeling). We do know that ^{18}F trifluoroethoxy groups can resist defluorination, as we observed with ^{18}F PS13 (Taddei et al. *ACS Chem. Neurosci.* 2021, 12, 517–530) where the ^{18}F -label is at the trifluoroethoxy position.

-Besides paraformaldehyde, are these methods developed in this manuscript also applicable to other aldehydes?

In our preliminary work, these methods are equally applicable to other aldehydes. However, that work is beyond the scope of this manuscript and will be communicated separately.

-What is the antimetabolic stability of the molecules with potential biological activity shown in this paper? It is suggested to evaluate the possibility of defluorination in vivo by PET of small animals.

Please see preceding response on the metabolic stability of trifluoroethoxy groups. PET experiments with some of the labeled compounds in this manuscript is beyond the scope of this present report. Detailed studies would be required on each prospective PET radiotracer. Stability might vary between such tracers.

REVIEWERS' COMMENTS

Reviewer #1 (Remarks to the Author):

The molar activity of ^{18}F is pretty low. At least, the molar activity of one ^{18}F final product should be tested. In most cases, the free ^{18}F ion could not be detected by HPLC in basic conditions.

All the other questions have been settled, and the details have been much improved. The manuscript was qualified for publication.

Reviewer #3 (Remarks to the Author):

My comments were fully addressed. Of course, this revised manuscript was recommended for publication in Nature Communication.

Response to Reviewers

We thank the Reviewers for thorough review of our manuscript on the *“Isotopologues of Potassium 2,2,2-Trifluoroethoxide for Applications in Positron Emission Tomography and Beyond”*.

We appreciate your positive review and the comments to improve the manuscript.

Here is our response to the reviewers how we addressed them. Matters needing attention by reviewers are highlighted in yellow; our responses to reviewers are in blue.

Reviewer: 1

Comments:

The molar activity of ^{18}F is pretty low. At least, the molar activity of one ^{18}F final product should be tested. In most cases, the free ^{18}F ion could not be detected by HPLC in basic conditions.

All the other questions have been settled, and the details have been much improved. The manuscript was qualified for publication.

We thank the reviewer for the positive comments on the manuscript to publish this work in Nature Communications.

We agree with the reviewer's comment on the low molar activity of ^{18}F -trifluoroethoxylation reactions. The experimentally measured molar activity of ^{18}F 1 of 1.3 GBq/ μmol is very low. However, this is not due to carrier dilution in the reaction used to prepare ^{18}F CF₃CH₂OK or in subsequent functionalization. For convenience, we used a literature method for producing the starting ^{18}F fluoroform, which gives low molar activity (van der Born et al. *Ang. Chem. int. Ed*, **2014**, 53, 11046-11050). The carrier dilution we observed was as expected for this method.

The main aim of the manuscript is to develop a new method for the transformation of radiolabeled fluoroform into highly reactive radiolabeled potassium 2,2,2-trifluoroethoxide (CF₃CH₂OK) synthon and demonstrate robust applications of this synthon in one-pot, two-stage 2,2,2-trifluoroethoxylation of both aromatic and aliphatic precursors. First, we have shown the robust applications of ^{11}C CF₃CH₂OK in ^{11}C -trifluoroethoxylation reaction. Then, we aimed to show that these ^{11}C -trifluoroethoxylation would readily and equally translate to ^{18}F -trifluoroethoxylation with ^{18}F fluoroform. This we demonstrated; use of ^{18}F fluoroform at higher molar activity would give correspondingly higher molar activity ^{18}F -trifluoroethoxylated products.

We clearly state and explain the reason for low molar activity of ^{18}F 1 in the final paragraph on page 7 of the manuscript, as follows: “The method of ^{18}F fluoroform synthesis that we used was one reported in the literature (48) and known to give low molar activity. We did not observe any significant release of fluoride ion in the production of ^{18}F potassium 2,2,2-trifluoroethoxide under basic conditions, as evidenced by absence of ^{18}F fluoride at the solvent front in the HPLC analysis of derived ^{18}F 2,2,2-trifluoroethanol (Supplementary Figure 18). Therefore, the molar activity of the starting ^{18}F fluoroform determines the molar activity of ^{18}F -labeled 2,2,2-trifluoroethoxy products.” One would expect higher molar activity of ^{18}F 1 if used higher molar activity ^{18}F fluoroform used in the production of ^{18}F CF₃CH₂OK.

Reviewer: 3

My comments were fully addressed. Of course, this revised manuscript was recommended for publication in Nature Communication.

We thank reviewer 3 for the recommendation to publish this work in Nature Communications.